# Rice False Smut Detection and Prescription Map Generation in a Complex Planting Environment, with Mixed Methods, Based on Near Earth Remote Sensing

**Fengnong Chen** [1,*] **, Yao Zhang** [1] **, Jingcheng Zhang** [1] **, Lianmeng Liu** [2] **and Kaihua Wu** [1]

1   College of Automation, Hangzhou Dianzi University, Hangzhou 310018, China; zhangyao@hdu.edu.cn (Y.Z.); zhangjcrs@hdu.edu.cn (J.Z.); wukaihua@hdu.edu.cn (K.W.)
2   China National Rice Research Institute, Hangzhou 310006, China; liulianmeng@caas.cn
*   Correspondence: fnchen@hdu.edu.cn; Tel.: +86-135-8880-5823

**Abstract:** Rice false smut is known as the cancer of rice. The disease is becoming increasingly prominent and is one of the major diseases in rice. However, prevention and treatment of this disease relies on "Centralized pesticide spraying". However, indiscriminate spraying leads to more pesticide residue, and impacts ecological and food safety. To obtain more objective results, different experimental planting forms are necessary. This study collected data at a complex planting environment based on "near earth remote sensing" using a frame-based hyperspectral device. We used mixed detection methods to differentiate between healthy rice and *U. virens* infected rice. There were 49 arrangements and more than 196 differentiation models between healthy and diseased rice, including 7 sowing data plots, 2 farm management types, and 23 pattern recognition methods. Finally, the real accuracy was mostly above 95%. In particular, with the increase of epoch and iteration, feature sequences based on deep learning could achieve better results; most of the accuracies were 100% with 100 epochs. We also found that differentiation accuracy was not necessarily correlated with the sowing dates and farm management. Finally, the detection method was verified according to the actual investigation results in the field. The prescription map of disease incidence was generated, which provided a theoretical basis for the follow-up precision plant protection work.

**Keywords:** rice false smut (*Ustilaginoidea virens*); complex plant environment; mixed discriminant methods; near earth remote sensing; detection; prescription map; precision plant protection

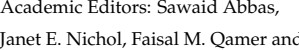

## 1. Introduction

Rice false smut (RFS), resulting from *Ustilaginoidea virens*, grows on rice grains and leads to heavy losses of rice yield in most major rice-producing areas. RFS was previously recorded as a minor disease of rice and considered a symbol of a good harvest in old times. In recent years, increasing occurrences of RFS have been reported in most major rice-growing regions throughout the world, such as China, India, and USA [1], causing chalkiness, and reducing "1000-grain weight" and seed germination (by up to 35%). In damp weather, the disease could be severe with losses reaching 25%. In India, a yield loss of 7–75% was observed [2]. Moreover, it is still viable in the soil and infects seedlings after planting [3]. Ustiloxins produced by *U. virens* pose as a serious hazard to human health and to the ecological safety of farmlands [4].

Suitable management practices need to be made to avoid the disease, in order to minimize direct economic loss. Breeding and utilization of a resistant cultivar is the most effective and economical way to control RFS disease and ensure the high yield of rice. Culture management will affect the incidence of RFS and early-planted rice has less RFS balls than late-planted rice; moreover, high nitrogen increases disease incidence. Chemical control is effective, for example, using fungicides with high efficiency, low toxicity, and low

residue is currently the best choice to control RFS disease, but it is often not environmentally-friendly. If the detection of RFS is more accurate in farmlands, plant protection will be more precise.

In the literature, many researchers have looked into the biological and chemical methods for RFS detection. For instance, Tang [5] developed a nested polymerase chain reaction (PCR)-based assay to detect *U. virens* using the genes of *U. virens* as specific targets. The study might be conducted to detect infection by *U. virens* at an early stage—to look into how to mitigate disease spread and to study the ecology. To evaluate disease severity, Dhua [6] presented a precise assessment method; a yield based on the florets and actual grains was simulated for the disease severity assessment of RFS disease. The above methods are relatively tedious. If a nondestructive and rapid method, e.g., remote sense, is used to detect RFS, it will be very convenient.

Lin [7] started with the pathogenesis of false smut bacteria and occurrence; he combined meteorological, agricultural, and remote sensing elements, and considered the meteorological factors conducive to false smut pathogen infection and dissemination, field management, rice varieties, nitrogen fertilizer, etc. From the perspective of bacteria pathogenesis and the occurrence of RFS, the study selected five indicators to simulate a forecast model of the RFS index (50 samples in 2 years) through the model verification; the results generally agreed with the actual situation, and the verification accuracy reached 83.32%.

Fewer studies on RFS have been based on near earth remote sensing. A large number of researchers have emphasized other crop diseases, such as satellite remote sensing for wheat Fusarium head blight [8], soybean sudden death syndrome [9], tobacco crop [10], rice bacterial leaf blight [11], soybean sudden death syndrome [12], near earth remote sensing for cucumber leaves in response to angular leaf spot disease [13], early disease in wheat fields [14], watermelon disease detection [15], rye leaf rust symptoms [16], paddy leaf disease [17], onion purple blotch [18], etc.

Among the literature, for satellite remote sensing research, Landsat imagery is free, but the spatial resolution is too low to accurately map smaller infestations. There are also satellites with pixel sizes of 5 m or less, such as GeoEye-1, Pleiades, WorldView-3 and -4, and GaoJing-1, or imagery with resolutions of 5–10 m, such as RapidEye, SPOT 6 and 7, and Sentinel-2, which are only suitable for macro-crop detection—near remote sensing imagery has a fine pixel size [19].

Although many crop diseases can be successfully detected and mapped using airborne or satellite imagery—understanding how to convert remote sensing data to practical prescription maps is still lacking. More research is needed to develop operational procedures for transforming image classification maps to applications maps. Each disease has its own characteristics and requires different procedures for detection and management.

For RFS detection, there are various influencing factors affecting RFS detection in near earth remote sensing, such as different sowing dates and farm management types. In this study, to obtain a more convincing and generalized model for RFS detection, we considered the sowing data and farm management types; 14 paddy fields were prepared, including 7 different sowing dates and 2 different farm management types. The data were obtained using a frame-based hyperspectral image device, based on near earth remote sensing; 23 different differentiation tradition models were built to obtain the most useful model. Spectral based deep learning was also used for detection. Finally, the selected model was verified by actual field investigation results. Our specific objectives were: (1) to ensure differentiation between healthy ears of rice (HER) and disease ears of rice (DER) at different sowing dates and farm management types. (2) To develop different models and get the most reliable model. (3) To develop an appropriate structure of the spectral deep learning model. (4) To verify the model by actual field investigation results; the disease prescription map was generated. The workflow of this study is shown in Figure 1.

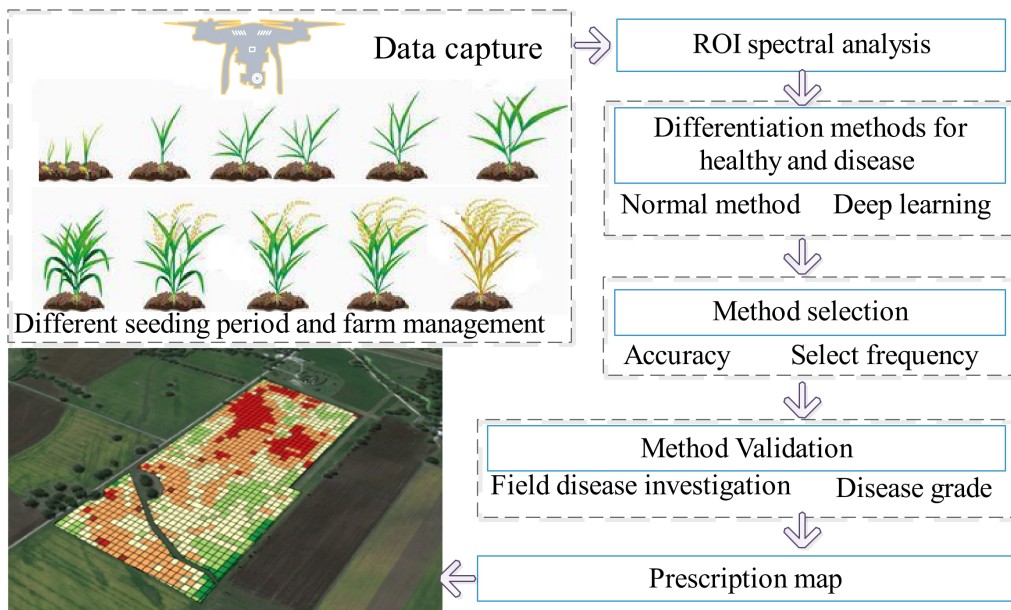

**Figure 1.** Workflow of this work.

## 2. Materials and Methods

### 2.1. Samples and Data Acquisition

The experiment fields were performed at the China National Rice Research Institute in Fuyang (120.2°E, 30.3°N, at an elevation of 11 m above sea level), Hangzhou, Zhejiang, in September 2017–2020. The research field covers over 500 hectares and includes 45,000 m² of professional research grounds; the soil is Ferric–Accumulic Stagnic Anthrosol [20]. The site is located in the middle and lower Yangtze Plain. The area is characterized by a subtropical monsoon climate with annual mean temperatures of 13–20 °C, ranging from 2 °C in January to 35 °C in July, and a mean annual precipitation of 1200–1600 mm, with approximately 80% falling between April and September [21].

The rice cultivar was Yongyou 12 [22]; this is a major rice cultivar with high yield for single-season rice in southeastern China. Rice plants were grown with 7 plots at different seed periods, and each plot was divided into 2 fields—nature growth (a pesticide absence treatment was used to induce the natural occurrence of RFS, NGT) and farm management (the field was set up as a reference plot and controlled by pesticide applications, FMT), as shown in Figure 2. There were a total of 14 different rice paddy fields; 10 data acquisition spots and DER were randomly selected in each field. HER were collected from the plants near the infested plants in the same data acquisition spot. Overall, 140 ears of rice samples were collected for DER and HER.

A spectrum from the ear of rice was obtained using S185 (Cubert GmbH, Ulm, Baden-Württemberg, Germany) from 12:00 to 14:00 under cloudless and windless weather. Before data collection, a black-and-white board was employed for radiation calibration of the S185. The hyperspectral image data were collected in the rice field with a hyperspectral frame-based camera, providing 137 channels, with a sampling interval of 4 nm in a spectral range of 450 to 950 nm. The full frame images were acquired by a silicium CCD chip with a sensor resolution of 1 megapixel, while the hyperspectral resolution was about 50 × 50 pixels with 12-bit (4096 DN) precision. A gray scale image with a resolution of 1000 × 1000 pixel was acquired. Accordingly, this camera's acquisition frequency was faster than the line scan hyperspectral camera. Finally, a 1000 × 1000 × 137 hyperspectral image was generated from the two files using the interpolation method.

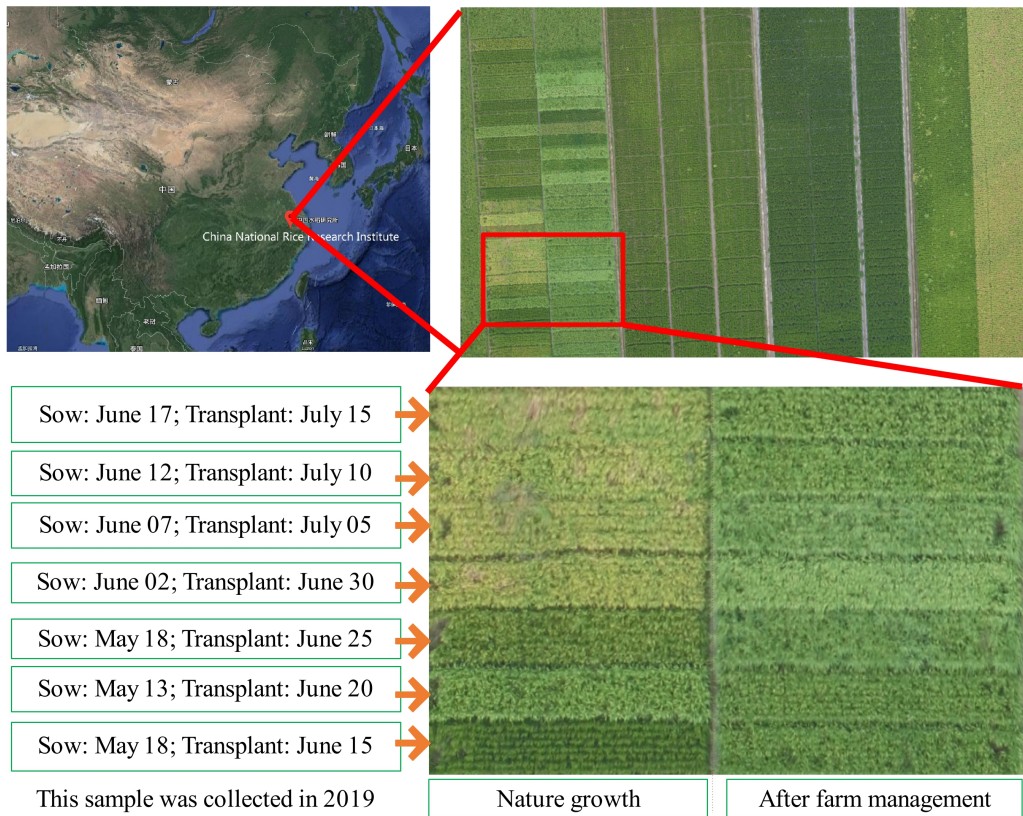

**Figure 2.** Location and details of test field.

The representative regions of interest (ROIs) reflectance spectra of samples in the wavelengths ranged from 400 to 950 nm, and were extracted from the healthy region and RFS regions. Each spectrum was obtained from a rectangular $3 \times 3$ pixel ROI. The average spectra are shown in Figure 3. Throughout the spectral region, the HER spectrum had a higher reflectance intensity than that of the RFS region. Thus, a single band image could discriminate between the healthy region and unhealthy regions by using a simple threshold (it can be seen from Figure 3 that the near-infrared spectral absorption of the diseased area is relatively large; the threshold can be set to 40 in this sample). However, the results showed that it was difficult to obtain a satisfactory result due to non-uniform light on the ear of rice. Furthermore, the spectra shown in Figure 3 do not account for all of the spatial variations, since the ear of rice had different angles. Subsequently, the dimensionality reduction method was used for the wavelengths (here, we used PCA to carry out a dimension reduction analysis; the details can be seen in Section 2.3).

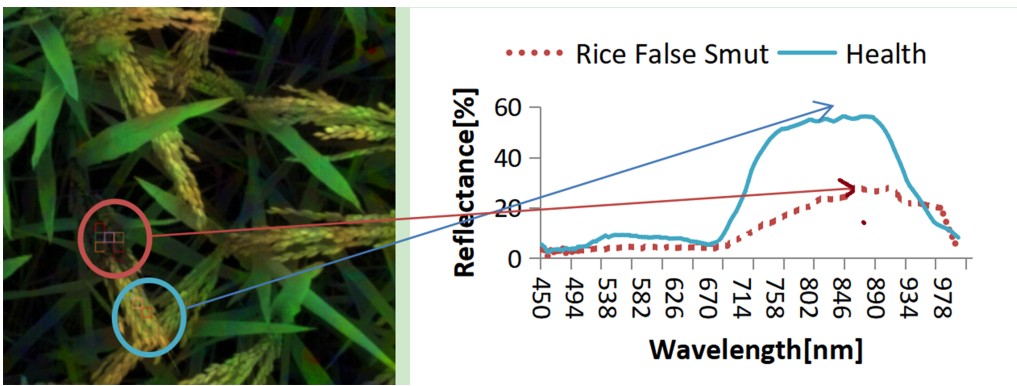

**Figure 3.** Image and spectral hyperspectral data in the healthy and RFS regions (typical and representative spectral data are shown here; the spectral trends of other samples are similar).

## 2.2. Data Analysis

The ROI data were obtained from CubePilot, the hyperspectral image device S185 with its own software (the data were collected in the same test site for 4 years, for at least twice a year. For example, data were collected twice in 2018, respectively, on 18 September and 8 October. In Figure 2, data were collected on 18 September 2018; rice was in the yellow-ripening stage. The weather was cloudy and windless). All data were analyzed using MATLAB 2019a (the MathWorks Inc., Natick, MA, USA) with an image processing toolbox (https://www.mathworks.com/help/stats/classification-learner-app.html, accessed on: 14 February 2020). In this study, several types of pattern recognition methods were applied for the data analysis. The details are seen at the following section. The flow of data analysis is shown in Figure 4.

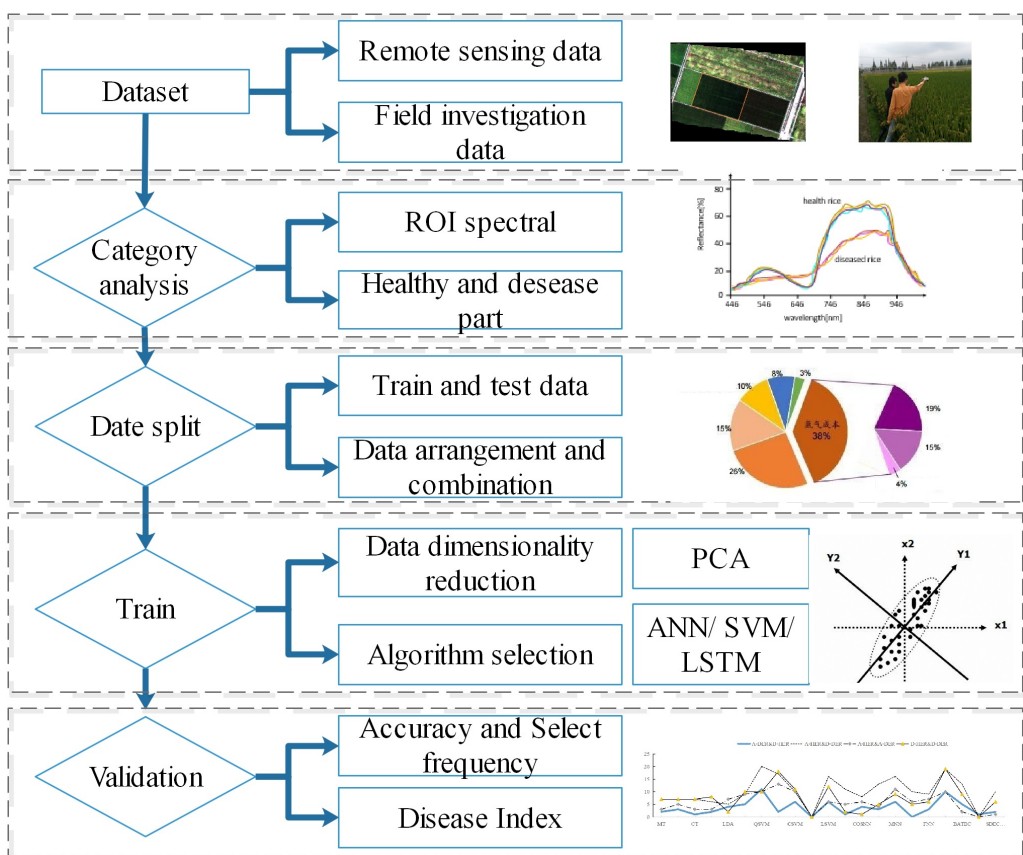

**Figure 4.** The flow of data analysis.

## 2.3. Differentiation Methods

To further study RFS estimation via detection models, sampling methods were used to produce calibration sets after data dimensionality reduction of the principal component analysis (PCA). In this study, the top-*n* principal component (PC) took up 95% of the total variance. A five-fold cross validation method was performed to protect against overfitting by partitioning the data set into folds and estimating the accuracy on each fold. Scaling the data sets in the study was standardized (we measured the distance of a data point from the mean, in terms of the standard deviation. $Z = (x - X)/S$: $x$ is data point, $X$ is mean, $S$ is standard deviation) to avoid attributes in greater numeric ranges dominating those in smaller numeric ranges and numerical difficulties during the calculation. The decision methods and descriptions are listed in Table 1. Comparisons of these methods have been described in many papers, e.g., retrieving the soybean leaf area index from UAV hyperspectral remote sensing using RF, ANN, and SVM regression models [23], analysis on change detection techniques for remote sensing [24], and the benefits of each approach in plant stress detection [25].

**Table 1.** Decision methods in the research.

| Classifier | Kernel Function | Code | Description |
|---|---|---|---|
| Decision trees | Complex tree | CT | Many leaves to make many fine distinctions between classes |
| | Medium tree | MT | Medium number of leaves for finer distinctions between classes |
| | Simple tree | ST | Few leaves to make coarse distinctions between classes |
| Discriminant analysis | Linear | LDA | Creates linear boundaries between classes |
| | Quadratic | QDA | Creates nonlinear boundaries between classes |
| Logistic regression | | LR | A popular classification algorithm for two classes |
| Support vector machines | Linear | LSVM | Creates linear boundaries |
| | Quadratic | QSVM | Creates nonlinear boundaries |
| | Cubic | CSVM | The solution is a piece-wise cubic |
| | Fine Gaussian | FSVM | Makes finely detailed distinctions between classes |
| | Medium Gaussian | MSVM | Kernel scale set to square root of P–sqrt(p), where P is the number of predictors. |
| | Coarse Gaussian | CGSVM | The same as above medium Gaussian, Kernel scale set to sqrt(P)*4 |
| nearest neighbors | Fine | FNN | The number of neighbors is 1 |
| | Medium | MNN | The number of neighbors is 10 |
| | Coarse | CNN | The number of neighbors is 100 |
| | Cosine | COSNN | Using a Cosine distance metric. The number of neighbors is 10 |
| | Cubic | CUBNN | Using a cubic distance metric. The number of neighbors is 10 |
| | Weighted | WNN | Using a distance weight. The number of neighbors is 10 |
| Ensemble classification | Boosted trees | BOTEC | AdaBoost, with decision tree learners |
| | Bagged trees | BATEC | Random forest Bag, with decision tree learners |
| | Subspace discriminant | SDEC | Subspace, with discriminant learners |
| | Subspace KNN | SKEC | Subspace, with nearest neighbor learners |
| | RUSBoost trees | RUSBEC | GentleBoost or LogitBoost, with decision tree learners, choose boosted trees and change to GentleBoost method |

Remote sensing data usually appear in the form of spectrum or sequences (data do not exist independently, before and after, or the whole sequence or data are associated); a long short-term memory (LSTM) network is more suitable for sequence data. This study classifies sequence data using the LSTM network in f MATLAB (https://ww2.mathworks.cn/discovery/lstm.html, accessed on 14 February 2021). Standard LSTM networks process sequences in chronological order, which ignore the future context. BiLSTM was chosen as the architecture because of the correlation between each sequence. Adaptive moment estimation (ADAM) was used as a solver for the training network, which uses a parameter update with an added momentum term. Other options include; MaxEpochs-600, three iterations per epoch; MiniBatchSize-27; learning rate: 0.001; sequence length-pad sequences in each mini-batch to have the same length as the longest sequence; shuffle–shuffle the training, and validation data once before training.

There were four different permutation comparison methods (NGT/DER, NGT/HER, FMT/DER, FMT/DER) among these testing data, namely HER and DER in the nature grown and management fields, respectively. There were 49 combinations of experiment plots (FMT1 vs. NGT 1, 2, ..., 7, FMT2 vs. NGT 1,2, ..., 7, ...), suggesting that there were a total of 196 differentiation results ($4 \times 49 = 196$) between FMT and HER. We used many permutations and combinations to take into account the diversity of the actual farmland.

### 3. Results and Discussion

*3.1. Comparisons between Different Test Fields*

Tables 2–5 shows the accuracy between DER and HER in the farm management of FMT and NGT. The name rules in the table are based on different test plots. For instance, 1A denotes the earliest and 7A the latest sowing plot with farm management. Likewise, 1D is the earliest and 7D is the latest sowing plot with nature growing. The remaining names

in table from 1A (D) to 7A (D) are consistent one-to-one matches between different sowing plots.

PCA was employed to reduce feature dimension, which kept enough components to explain the 95% variance in this study. The maximum accuracy could be obtained by more than one method. The best methods were statistically analyzed according to the selected frequency (Figure 4). In Table 2; the accuracy ranged from 87 to 100%, mostly above 95%.

**Table 2.** The accuracy (%) between DER in FMT and HER in NGT.

|        | 1D-HER | 2D-HER | 3D-HER | 4D-HER | 5D-HER | 6D-HER | 7D-HER |
|--------|--------|--------|--------|--------|--------|--------|--------|
| 1d-HER | 87.8   | 94     | 95     | 96.4   | 91.1   | 92.3   | 87     |
| 2d-HER | 92.3   | 97.6   | 93.1   | 97.9   | 92.2   | 92.7   | 87     |
| 3d-HER | 93     | 96.9   | 96     | 94.4   | 94     | 92     | 91     |
| 4d-HER | 96.1   | 97.9   | 95.5   | 96.7   | 96.5   | 92.7   | 93.3   |
| 5d-HER | 90.2   | 97.1   | 96.3   | 94.1   | 95.2   | 88.1   | 89.4   |
| 6d-HER | 98.4   | 100    | 98.1   | 98.4   | 88.9   | 95.1   | 94.9   |
| 7d-HER | 99.5   | 98.7   | 98.5   | 100    | 99.6   | 96.4   | 97     |

Note: HER—healthy ears of rice; DER—disease ears of rice; NGT—nature growth treatment; FMT—farm management treatment. The abbreviations of the following three tables are also consistent with this table.

**Table 3.** The accuracy (%) between DER in NGT and HER in FMT.

|        | 1D-HER | 2D-HER | 3D-HER | 4D-HER | 5D-HER | 6D-HER | 7D-HER |
|--------|--------|--------|--------|--------|--------|--------|--------|
| 1d-DER | 99.4   | 100    | 100    | 100    | 98.8   | 99.2   | 98.2   |
| 2d-DER | 99.5   | 100    | 99.3   | 98.7   | 98.6   | 99.7   | 98.5   |
| 3d-DER | 99.6   | 99.6   | 99.3   | 100    | 100    | 100    | 99.7   |
| 4d-DER | 97     | 93.8   | 97.1   | 98.5   | 99.3   | 94.5   | 97     |
| 5d-DER | 99.7   | 99.7   | 99.8   | 100    | 97.4   | 99.5   | 98.8   |
| 6d-DER | 93.8   | 90.8   | 96.6   | 96.7   | 97.3   | 91     | 90.7   |
| 7d-DER | 90.4   | 91.4   | 93.3   | 95.9   | 87.3   | 94.9   | 92     |

**Table 4.** The accuracy (%) between DER and HER in FMT.

|        | 1D-HER | 2D-HER | 3D-HER | 4D-HER | 5D-HER | 6D-HER | 7D-HER |
|--------|--------|--------|--------|--------|--------|--------|--------|
| 1A-DER | 98.6   | 96.7   | 99     | 98.7   | 95.8   | 98.1   | 97.6   |
| 2A-DER | 98.9   | 99     | 100    | 99.2   | 100    | 98.9   | 98.4   |
| 3A-DER | 99.1   | 96.3   | 99     | 98.7   | 98     | 97.1   | 97.2   |
| 4A-DER | 92.5   | 98.9   | 98.8   | 100    | 96.4   | 99.2   | 97     |
| 5A-DER | 99.3   | 98.3   | 98.8   | 100    | 94.4   | 98.4   | 97.1   |
| 6A-DER | 95.9   | 94.6   | 96     | 94.9   | 92.5   | 95.9   | 94.8   |
| 7A-DER | 97.8   | 95.5   | 96     | 96.2   | 91.6   | 94     | 92.5   |

**Table 5.** The accuracy (%) between DER and HER in NGT.

|        | 1D-HER | 2D-HER | 3D-HER | 4D-HER | 5D-HER | 6D-HER | 7D-HER |
|--------|--------|--------|--------|--------|--------|--------|--------|
| 1d-DER | 97.4   | 96.7   | 96.3   | 98.2   | 97.8   | 100    | 100    |
| 2d-DER | 98     | 96.4   | 97.7   | 98.7   | 97.2   | 96.8   | 100    |
| 3d-DER | 99.1   | 99.6   | 100    | 97.8   | 99.6   | 100    | 100    |
| 4d-DER | 97.5   | 93.3   | 96.1   | 92.4   | 95.5   | 99     | 97.1   |
| 5d-DER | 96.8   | 96.2   | 97.5   | 98.7   | 94.8   | 99.5   | 100    |
| 6d-DER | 96.5   | 84.2   | 84.6   | 89.7   | 79.1   | 92.6   | 96.4   |
| 7d-DER | 96.8   | 91.5   | 91.5   | 86.6   | 88.8   | 94.1   | 89.1   |

### 3.2. Accuracy between Different Farm Managements of FMT and NGT

Figure 5 shows the results of the differentiation between FMT and NGT. The plot covers 196 results with different arrangements. Box-plots for the sample set involved the representatives of different sowing periods and different farm managements of rice

growth. In Figure 4, the horizontal ordinate represents the table results different from that of Tables 2–5 above. From the box-plots, the accuracies ranged from 92 to 99%.

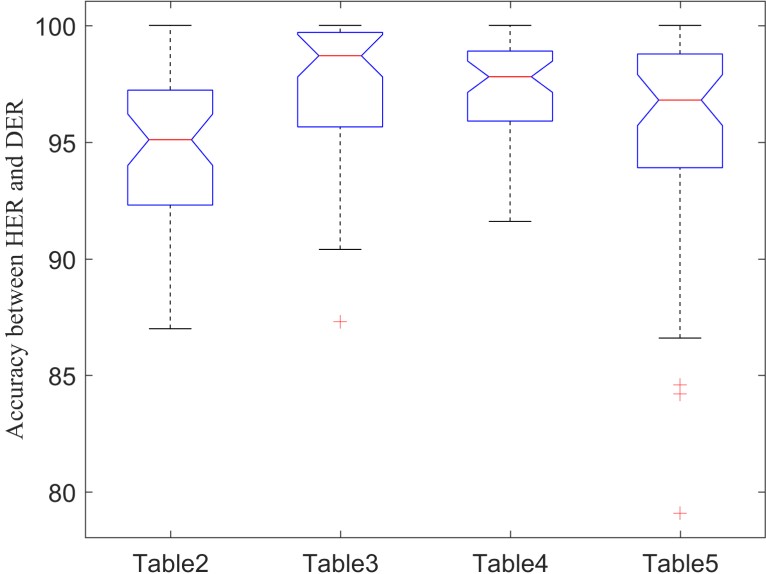

**Figure 5.** Different comparison methods between FMT and NGT (Table 2 represent the classification accuracies of all combinations of sowing dates and farmland management methods between DER in FMT and HER in NGT. Similarly, Table 3 is the accuracy between DER in NGT and HER in FMT; Table 4 is the accuracy between DER and HER in FMT; Table 5 is the accuracy between DER and HER in NGT).

The differentiation methods between DER and HER among the different farm managements and sowing dates were analyzed. Among the arrangements, WNN was the most frequently used differentiation method for RFS analysis. Many researchers [26] have already estimated various differentiation models; moreover, there are different opinions about which model is the most appropriate for DER and HER differentiation. This study used 23 discrimination methods, a very common algorithm at present, to detect all possible combinations, to ensure the effectiveness of the differentiation results obtained. Those combinations not only covered different farm management types, but different sowing periods. For each combination, there will be more than one method to get the highest accuracy, so the selection frequencies of all discrimination methods were counted, as shown in Figure 6. It is suggested that QSVM, WNN, and LSVM were the top-three highest accuracies for A-DER and D-HER; QSVM, WNN, and FSVM were the top-three for A-HER and D-DER; FSVM, MNN, and QSVM were the top-three for A-HER and A-DER; WNN, FSVM, and LSVM were the top-three for D-HER and D-DER. Generally, WNN and QSVM were the most selective and, therefore, these two methods could be considered the best among traditional models.

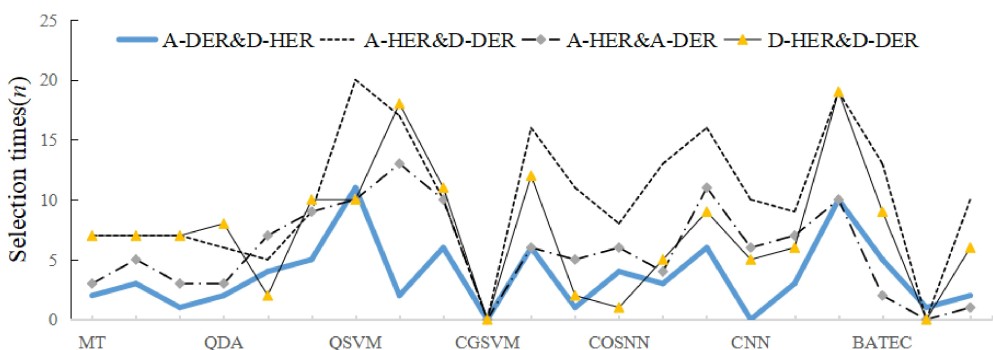

**Figure 6.** Selected frequencies of discrimination methods for plot combinations (A and D, respectively, represent plots with two different management, with a total of four combination modes. The abbreviation of abscissa represents the different detection methods, and the ordinate value represents the number of times to obtain the highest accuracies in multiple combinations).

### 3.3. Spectral-Based Deep Learning

Spectral-based deep learning required approximately 3 min to assess these datasets. Compared to deep learning for slide-images, the elapsed time was shorter (the platform was based on a Lenovo P53, CPU Intel Core i5-9400H 2.50 GHz, RAM 32 GB ). If there are special requirements for time, this method is feasible. For images, it will take days to train. Therefore, spectral-based deep learning greatly improves the efficiency of detection.

According to the detection results using LSTM, most accuracies achieved over 95%. The real accuracies ranged from 90 to 100%. The accuracy increased significantly from the beginning of training to the 500 iteration. Then it slowly rose. As for loss value—it decreased gradually with the number of iterations. The smoothed value was less than 0.1. In Table 6 below, DN is DER in NGT, DF is DER in FMT, HN is HER in NGT, and HF is HER in FMT. The line is 14 kinds of disease samples and the row is 14 kinds of health samples, under two different farm management types.

Each crop (different from industrial products) is different; thus, the diversity of rice planting should be considered in disease detection. In this study, we considered different the planting methods of rice and attempted to use a variety of detection methods. The main purpose was to make the research results closer to the actual production and to have better popularization.

According to the analysis results, it can be found that the accuracies nearly ranged from 92 to 99%. Generally, better results could be obtained, which is irrelevant to the management mode and planting date.

Data have a great impact on the results, especially in rice fields. Moreover, one must consider the impact of data collection methods and data inconsistency caused by different manufacturers. Therefore, in order to ensure the consistency of data, it is necessary to avoid the influences caused by the change of sunlight through radiation correction devices, and avoid the impact caused by the inconsistency of equipment manufacturers and personal use habits.

### 3.4. Method Validation

Grading of the rice disease index divided RFS into 5 grades, of which, 0 indicated no diseases, 1 was the least, and 5 was the most serious [27]. The incidence rate of RFS in the unit area was used as the classification standard of the disease grades. The specific description was as follows:

Grade 0: The prevalence rate of RFS was 0;
Grade 1: The prevalence rate of RFS was under 5%(include 5%);
Grade 2: The prevalence rate of RFS was under 10%(include 10%);
Grade 3: The prevalence rate of RFS was under 20%(include 20%);
Grade 4: The prevalence rate of RFS was under 50%(include 50%);

Grade 5: The prevalence rate of RFS was more than 50%.

According to this standard, the disease index was calculated according to the field disease investigation results, the disease index was as follows:

$$DI = \frac{\sum x_i \times n_i}{N \times k} \tag{1}$$

where, $DI$ is the disease index; $x_i$ is the number of diseased panicles at all levels (number/plot); $n_i$ is the representative value at all levels; $N$ is the total number of panicles investigated (number/plot); $k$ is the highest representative value. In order to facilitate the calculation, experimental fields were finally divided into several grades according to the size of the disease index.

The disease grade of RFS, as the control group, was obtained through a field investigation. In this study, the RFS with different diseased grades were distinguished and counted according to the rice disease index. The incidence index of each area was calculated (Table 6). The incidence grade was divided into four grades 1, 3, 5, and 7, according to the disease index. The conditions corresponding to the disease index were $DI < 1$, $1 < DI < 3$, $3 < DI < 5$ and $DI > 5$. The classification of grades in this study was mainly based on the distribution gradient of the disease index.

**Table 6.** Disease index based on the field investigation of RFS in different experimental areas under five diseased grades.

| Field Label | Number of Diseased Plants | | | | | | DI |
|---|---|---|---|---|---|---|---|
| | 0 | 1 | 2 | 3 | 4 | 5 | |
| 1A | 500 | 0 | 0 | 0 | 0 | 0 | 0 |
| 1B | 500 | 0 | 0 | 0 | 0 | 0 | 0 |
| 1C | 499 | 1 | 0 | 0 | 0 | 0 | 0.022 |
| 1D | 498 | 1 | 1 | 0 | 0 | 0 | 0.089 |
| 2A | 499 | 0 | 0 | 1 | 0 | 0 | 0.111 |
| 2B | 498 | 0 | 2 | 0 | 0 | 0 | 0.133 |
| 2C | 500 | 0 | 0 | 0 | 0 | 0 | 0 |
| 2D | 499 | 0 | 0 | 0 | 0 | 0 | 0 |
| 3A | 500 | 0 | 0 | 0 | 0 | 0 | 0 |
| 3B | 498 | 0 | 2 | 0 | 0 | 0 | 0.133 |
| 3C | 499 | 0 | 1 | 0 | 0 | 0 | 0.067 |
| 3D | 498 | 4 | 3 | 0 | 0 | 0 | 0.286 |
| 4A | 469 | 16 | 7 | 5 | 2 | 1 | 1.889 |
| 4B | 482 | 10 | 2 | 4 | 2 | 0 | 1.111 |
| 4C | 474 | 16 | 6 | 2 | 2 | 0 | 1.289 |
| 4D | 469 | 16 | 7 | 5 | 2 | 1 | 1.889 |
| 5A | 481 | 10 | 1 | 6 | 2 | 0 | 1.267 |
| 5B | 433 | 32 | 20 | 14 | 1 | 3 | 4.33 |
| 5C | 423 | 24 | 19 | 15 | 8 | 6 | 5.971 |
| 5D | 363 | 59 | 27 | 24 | 13 | 14 | 10.6 |
| 6A | 560 | 19 | 9 | 8 | 5 | 1 | 2.399 |
| 6B | 515 | 19 | 16 | 9 | 2 | 1 | 2.669 |
| 6C | 513 | 12 | 10 | 8 | 4 | 0 | 2.234 |
| 6D | 191 | 10 | 2 | 10 | 4 | 0 | 4.813 |
| 7A | 431 | 69 | 23 | 24 | 2 | 4 | 6.188 |
| 7B | 373 | 59 | 33 | 40 | 11 | 12 | 11.43 |
| 7C | 429 | 45 | 9 | 15 | 2 | 1 | 3.77 |
| 7D | 129 | 41 | 8 | 13 | 4 | 9 | 13.017 |

Note: Groups *A, B, and C* used different pesticides to suppress diseases and insect pests, and the test field in group *D* grew naturally as a control group without any disease suppression treatment. The field label is consistent with that in the section of comparisons between different test fields, but there are four different field plots.

According to the detection method above (top-three highest accuracies and LSTM), we compared the results with the actual field investigation data, and the results were basically

consistent. Figure 7 is a prescription map of RFS detection in the experimental field. We not only carried out disease detection, but we also subdivided the disease grade. Deep green indicates less disease, red indicates serious disease, and light green and light red are between the two.

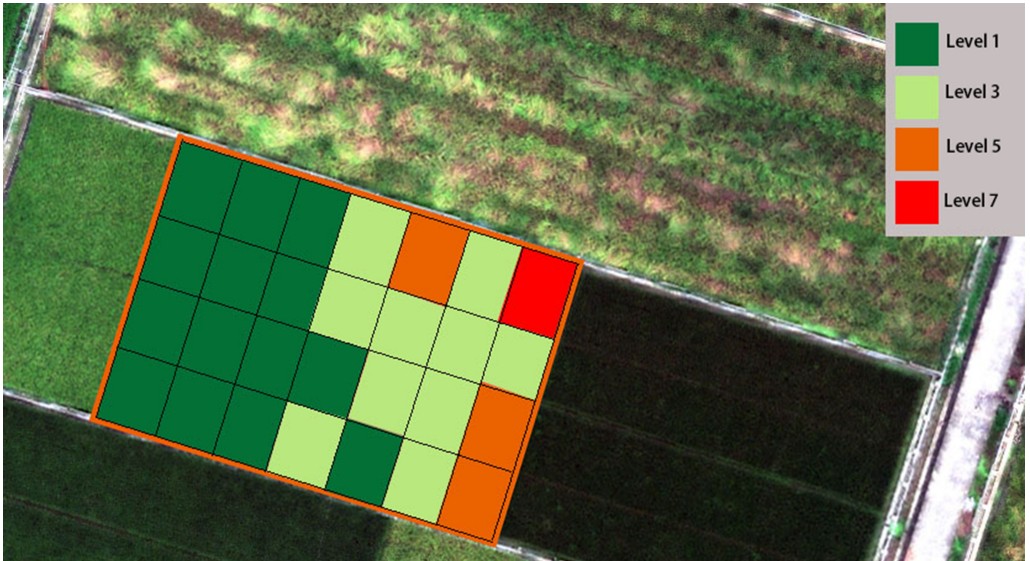

**Figure 7.** Prescription map of RFS detection in the experimental field.

Due to the large number of rice plants, we cannot guarantee that each disease area can be detected. In the process of near earth remote sensing data acquisition, we should make the resolution appropriate. If the data collected by UAV are too close to the ground and the wind force of UAV is too strong, the rice canopy fluctuates too much and better data cannot be obtained. If they are too far away from the rice plant, it is impossible to get a better resolution.

## 4. Conclusions

This study considered several combinations of rice plant forms, which covered different planting types and management methods. Of those samples, the most convenient method of the submitted algorithms was based on deep convolutional neural networks. As for traditional methods, the important step involves the features extracted; therefore, different statistical and structural features were extracted, combined with widely used supervised classifiers.

The real accuracy was mostly above 95%. Among these models, support machines and nearest neighbors with different kernel functions were generally better. WNN and QSVM were the most frequency selected methods. Moreover, there was no obvious characteristic showing which arrangement was better. Considering so many factors, the differentiations between healthy and diseased rice plants turned out to be more reliable.

According to these methods, the disease prescription map of RFS could be produced, which provides a theoretical basis to take corresponding control measures in the future.

Although this paper has "reference significance" in the detection of RFS, there were still some limitations.

(1) Near earth remote sensing was used to detect and map RFS in this study; early detection is still a challenge, in most cases, this delayed detection may be early enough to reduce further damage with certain measures for RFS; for others, it may be too late to stop the infection for the current growing season.

(2) Rice planting is affected by many factors; only the two most important factors for the incidence of RFS were considered in this study.

(3) This work was carried out in China, with a typical subtropical climate, and involving a certain popularization. If the same experiment is carried out in other rice producing areas, there may be differences.

(4) In addition, compared with the convenience of satellite data, image acquisition with UAV is a technical work, especially in preventing the damage of very expensive hyperspectral equipment. Unlike satellite data, there are some uncertainties in the consistency of near ground remote sensing data.

**Author Contributions:** F.C. writing, supported for the project; Y.Z. scrub and maintain research data; J.Z. design of methodology; L.L. and K.W. Provision of study materials. All authors have read and agreed to the published version of the manuscript.

**Funding:** This work was supported by the Zhejiang Province Key Research and Development Program (grant number: 2021C02011) and the Zhejiang Province Public Welfare Technology Application Research Project (grant number: LGN18F030002).

**Institutional Review Board Statement:** Not applicable.

**Informed Consent Statement:** Not applicable.

**Data Availability Statement:** Not applicable.

**Acknowledgments:** We thank the China Rice Institute for providing the experimental site and relevant samples.

**Conflicts of Interest:** The authors declare no conflict of interest.

## Abbreviations

The following abbreviations are used in this manuscript:

| | |
|---|---|
| *U. virens* | *Ustilaginoidea virens* |
| RFS | rice false smut |
| PCR | polymerase chain reaction |
| GA | genetic algorithm |
| LDA | linear discriminant analysis |
| HER | healthy ears of rice |
| DER | disease ears of rice |
| NGT | nature growth treatment |
| FMT | farm management treatment |
| ROI | region of interest |
| PCA | principal component analysis |
| BiLSTM | bidirectional long short-term memory |
| ADAM | adaptive moment estimation |
| DI | disease index |

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
