# Peer review of "Rice False Smut Detection and Prescription Map Generation in a Complex Planting Environment, with Mixed Methods, Based on Near Earth Remote Sensing"

_remotesensing, doi:10.3390/rs14040945_

Round 1

Reviewer 1 Report

Dear Authors,

Thank you for submitting to Remote Sensing Journal. This work seems to be interesting but there are serious flaws which need to be revised for publication.

  1. There seems to be many objectives in this paper so a flowchart have to be drawn to clearly depict the work in  a more clearer way.
  2. Separate flowchart should be drawn exclusively for machine learning models used in this study.
  3. Introduction seems to be incomplete. Please improve the introduction.
  4. State-of-the-art also have to be improved.
  5. There are many acronyms for which a table of nomenclature can be created.
  6. Please modify the resolution of hyperspectral image in terms of SI units.

Thank you.

Author Response

Thank you very much for your comments. All your suggestions are very useful. I made changes in the manuscript. Please take a look again. If you feel there are still problems, please tell me. I will try my best to make changes. Thank you.

  1. There seems to be many objectives in this paper so a flowchart have to be drawn to clearly depict the work in  a more clearer way.

OK,  I drawn the flowchart in the manuscript, please check it . What do you think? I'll revise it if you don't think it's OK.

  1. Separate flowchart should be drawn exclusively for machine learning models used in this study.

OK,  I drawn the flowchart in the manuscript.

  1. Introduction seems to be incomplete. Please improve the introduction.

I checked the relevant literature and improve it. Please have a look again.

  1. State-of-the-art also have to be improved.

OK, I have made changes.

  1. There are many acronyms for which a table of nomenclature can be created.

I've added it to the end of the text.

  1. Please modify the resolution of hyperspectral image in terms of SI units.

 OK, the current images are all 300 dpi.

Reviewer 2 Report

Please find below comments on your article.

Overall, I think this makes a nice contribution, but I think there are way to broaden the analysis for a wider audience, and provide more informed presentation of the data.

General comments

C1. The main concern is that the novelty of the research is not fully clear. If such novelty is not clearly highlighted, the risk is that the manuscript looks more a simple case study rather than a research paper.

C2. What are the major contributions and limitations of this study? should be carefully discussed in the discussion section.

C3. To the international readers who are engaged in the rice false smut, cropland management, what can be learned from this study?

C4. What are the advantages and disadvantages of this study?

C5. Authors could also emphasize particular strengths and limitations of the study for potential applications of their method in other regions, different types of crops.

C6. All titles of tables and figures have periods. Check the guide for authors.

*The answer to these questions should be reflected in the manuscript.*

Specific comments

Line 21: Scientific names are in italics. Change.

Line 97: Change "," to "."

Line 97: A complete map of the study area is required. See: https://doi.org/10.3390/rs13234944 , https://www.mdpi.com/2072-4292/14/2/307

Figure 1. Remove the outer gray box from the figure. What dates does the historical data correspond to?

Figure 2. What was the make and model of the UAV and hand-held capture?

Line 129: What is that simple threshold?

Line 132: What was the method of dimensionality reduction?

Figure 3. Clarify that this is an example and does not take into account all spatial variation.

Line 135-136: Provide day, month, year and access link.

Line 136-137: Provide program access link.

Line 144-146: How was the data scaling done?

Line 148: Mention that the methods are described in a large number of manuscripts, but only cite one. Add more sources.

Line 156-157: On what program or platform was this method used? Provide access link.

Table 2-5. Add in a table footnote what the abbreviations mean.

Line 205: What was the significance level?

Figure 4. Tables 3, 4 and 5 appear to have a very similar mean value, are they significantly different? Place an appropriate symbology for the boxplot.

Figure 5. Remove the title inside the figure "selected methods". Add what all the initials of the methods mean.

Line 222-223: In which software do we need to discuss the speed of data processing? Does this align with the objectives?

Line 267: Change "," to ".".

Table 6 Footnote: Delete the "..."

Line 291-End of discussion: Please, look at your Discussion, is there a real comparisons to other researchers of your results?.

Author Response

Thank you very much for your very useful comments. I made changes in the manuscript. Please tale a look again. If you feel there are still problems, please tell me. I will try my best to make changes. 

C1. The main concern is that the novelty of the research is not fully clear. If such novelty is not clearly highlighted, the risk is that the manuscript looks more a simple case study rather than a research paper.

There is not much research on Rice False Smut at present, so I did this work. At the same time, I also considered different planting dates and management methods, which is relatively more targeted. Of course, there are still many areas to be improved, and I hope you will criticize and correct next.

C2. What are the major contributions and limitations of this study? should be carefully discussed in the discussion section.

OK, it has been added at the end of the text

C3. To the international readers who are engaged in the rice false smut, cropland management, what can be learned from this study?

OK, I have explained it at the end of the manuscript.

C4. What are the advantages and disadvantages of this study?

OK, I have explained it in the manuscript.

C5. Authors could also emphasize particular strengths and limitations of the study for potential applications of their method in other regions, different types of crops.

OK, I have explained it at the end of the manuscript.

C6. All titles of tables and figures have periods. Check the guide for authors.

OK, I have revised it according to your requirements.

*The answer to these questions should be reflected in the manuscript.*

Specific comments

Line 21: Scientific names are in italics. Change.

OK, I have revised it according to your requirements.

Line 97: Change "," to "."

OK, I have revised it according to your requirements.

Line 97: A complete map of the study area is required. See: https://doi.org/10.3390/rs13234944 , https://www.mdpi.com/2072-4292/14/2/307

OK, I have revised it according to your requirements.

Figure 1. Remove the outer gray box from the figure. What dates does the historical data correspond to?

Some reviewers suggested deleting this figure and describing it directly in words.

Figure 2. What was the make and model of the UAV and hand-held capture?

I've redone the drawing now, because several reviewers have put forward a lot of opinions on the drawing

Line 129: What is that simple threshold?

I explained in the manuscript.

Line 132: What was the method of dimensionality reduction?

I explained in the manuscript.

Figure 3. Clarify that this is an example and does not take into account all spatial variation.

OK, I have revised it.

Line 135-136: Provide day, month, year and access link.

OK, I have provided the access link.

Line 136-137: Provide program access link.

OK, I have provided the link.

Line 144-146: How was the data scaling done?

OK, I have added it in the text.

Line 148: Mention that the methods are described in a large number of manuscripts, but only cite one. Add more sources.

OK, I have add relevant reference.

Line 156-157: On what program or platform was this method used? Provide access link.

OK, I have provided the link.

Table 2-5. Add in a table footnote what the abbreviations mean.

OK, I have added the footnote in the text.

Line 205: What was the significance level?

The significance level is statistical, and classification are two different directions. This significance analysis is really not done here. I'm also thinking about how to add it.

Figure 4. Tables 3, 4 and 5 appear to have a very similar mean value, are they significantly different? Place an appropriate symbology for the boxplot.

Yes, it's a combination of several ways. It's not easy to describe it in one table. But I've put footnotes on it.

Figure 5. Remove the title inside the figure "selected methods". Add what all the initials of the methods mean.

OK, it has been changed according to your requirements.

Line 222-223: In which software do we need to discuss the speed of data processing? Does this align with the objectives?

I added the model of the platform.

Line 267: Change "," to ".".

OK, I have revised it.

Table 6 Footnote: Delete the "..."

OK, I have revised it according to your requirements.

Line 291-End of discussion: Please, look at your Discussion, is there a real comparisons to other researchers of your results?.

Yes, I explained it in the text.

Reviewer 3 Report

Line 69: “For plant recognition, tradition models mainly concentrate on two steps: feature selection and classifier training.”

The Introduction should be centered on crop pest and disease monitoring. The methods related to plant recognition are not necessary to be mentioned in this section, and the paragraph is not supported by literature, the above content needs to be revised.

Line 103: “Figure 1. Average climatic conditions according to historical data in the research field”

For the meteorological conditions of the study area only a textual description is sufficient, Figure 1 is proposed to be deleted.

Line 103: “Figure 3. Image and spectral of hyperspectral data in health and RFS parts”

The legend of Rice False Smut in this figure is a dotted line, but it is not obvious in the figure, so we suggest to modify it.

Line 172: “Options for training deep learning neural network”

This part introduces the details of deep learning neural network. However, on the one hand, the parameters involved and the tuning process are not specific enough. On the other hand, this part is purely algorithmic, and it is suggested to introduce the details of the method in conjunction with RFS remote sensing monitoring to enhance the readability of this part.

Line 189: “3.1. Comparisons between different test fields”

Since the occurrence of RFS is related to the growth stage of rice, it is suggested that the authors write out the growth stage of rice in all experimental fields with different rice planting times in this section to analyze the accuracy of RFS monitoring under different growth stage.

Line 209: “Figure 4. Different comparison methods between FMT and NGT”

The meaning of the figure is not clear enough and readability is not strong, we suggest readers to strengthen the introduction of the figure.

Line 220: “Figure 5. Selected frequency of discrimination method for those plot combination”

The graph is missing names and units for the vertical coordinate.

Line 277: “Figure 6. Prescription map of RFS detection in the experimental field”

This paper mainly analyzed the spectral differences between diseased and healthy samples under the same pixel window and developed monitoring models. However, an experimental field contains many rice plants, how to obtain the RFS monitoring results of all experimental fields based on small-scale spectral features is not detailed in the text, which is not very readable, and the authors are suggested to strengthen the descriptions of this part.

Author Response

Thank you very much for your comments. All your suggestions are very useful. I made changes in the manuscript. Please tale a look again. If you feel there are still problems, please tell me. I will try my best to make changes. Thank you.

Line 69: “For plant recognition, tradition models mainly concentrate on two steps: feature selection and classifier training.”

The Introduction should be centered on crop pest and disease monitoring. The methods related to plant recognition are not necessary to be mentioned in this section, and the paragraph is not supported by literature, the above content needs to be revised.

Yes, I rewrite the introduction, please take a loot again.

 Line 103: “Figure 1. Average climatic conditions according to historical data in the research field”

For the meteorological conditions of the study area only a textual description is sufficient, Figure 1 is proposed to be deleted.

OK, it has been modified according to your requirements. Indeed, this figure is of little significance. I have deleted it.

Line 103: “Figure 3. Image and spectral of hyperspectral data in health and RFS parts”

The legend of Rice False Smut in this figure is a dotted line, but it is not obvious in the figure, so we suggest to modify it.

OK, it has been modified according to your requirements.

 Line 172: “Options for training deep learning neural network”

This part introduces the details of deep learning neural network. However, on the one hand, the parameters involved and the tuning process are not specific enough. On the other hand, this part is purely algorithmic, and it is suggested to introduce the details of the method in conjunction with RFS remote sensing monitoring to enhance the readability of this part.

This section is really not well. I wrote it again.

 Line 189: “3.1. Comparisons between different test fields”

Since the occurrence of RFS is related to the growth stage of rice, it is suggested that the authors write out the growth stage of rice in all experimental fields with different rice planting times in this section to analyze the accuracy of RFS monitoring under different growth stage.

Yes, this proposal is to the point. I have only two time points for data collection in the experimental field, that is, about a week before and after the onset of rice false smut. It would be better to detect the whole rice growth cycle, which can also carry out disease early warning.

 Line 209: “Figure 4. Different comparison methods between FMT and NGT”

The meaning of the figure is not clear enough and readability is not strong, we suggest readers to strengthen the introduction of the figure.

I added a note at the end of table.

 Line 220: “Figure 5. Selected frequency of discrimination method for those plot combination”

The graph is missing names and units for the vertical coordinate.

OK, I've revised it.

 Line 277: “Figure 6. Prescription map of RFS detection in the experimental field”

This paper mainly analyzed the spectral differences between diseased and healthy samples under the same pixel window and developed monitoring models. However, an experimental field contains many rice plants, how to obtain the RFS monitoring results of all experimental fields based on small-scale spectral features is not detailed in the text, which is not very readable, and the authors are suggested to strengthen the descriptions of this part.

OK, I've revised it and explained in the text.

Round 2

Reviewer 1 Report

Dear Authors,

Thank you for the modifications and the article seems to be in order now. Hope the article gets published convincing other reviewers too.

Thank you.